# Colon mucosal proteomics of ankylosing spondylitis versus gut inflammation

**Miao Cheng**[1,2☯], **Siqi Xiao**[1,2☯], **Shaer Kayi**[1,2☯], **Yujie Guan**[3], **Yingxin Liu**[1,2], **Jianmei Chen**[1,2], **Hua Chen**[4]*, **Lei Wang**[1,2]*, **Xiaojin He**[1,2]*

1 The Affiliated Hospital of Nanjing University of Chinese Medicine, Nanjing, China, 2 Jiangsu Provincial Hospital of Traditional Chinese Medicine, Nanjing, China, 3 Suzhou Hospital of Integrated Traditional Chinese and Western Medicine, Suzhou, China, 4 School of Nursing, Nanjing University of Chinese Medicine, Nanjing, China

☯ These authors contributed equally to this work.

* chenhua@njucm.edu.cn (HC); wonglay1989@163.com (LW); 260632@njucm.edu.cn (XH)

## Abstract

### Objective

Ankylosing spondylitis (AS) patients often present with microscopic signs of gut inflammation. We used proteomic techniques to identify the differentially expressed proteins (DEPs) in the colon tissues of patients with AS and patients with gut inflammation, and then used investigated the influence of NMRAL1 protein on inflammatory cytokines to explore its potential role in the pathogenesis of AS and gut inflammation.

### Methods

Colonic mucosal tissues were collected from four different groups: healthy individuals (group A), patients with gut inflammation only (group B), patients with AS only (group C), and patients with AS combined with gut inflammation (group D). A total of 20 samples were processed for proteomic analysis, wherein proteins were extracted using SDT lysis, followed by separation via sodium dodecyl sulfate–polyacrylamide gel electrophoresis (SDS-PAGE). The proteins were digested using the filter-aided sample preparation (FASP) method and then analyzed using a timsTOF Pro mass spectrometer. The resulting peptide data were used to identify differentially expressed proteins (DEPs) across the different groups. To further explore the inflammation-related function of NMRAL1 protein, the murine monocyte/macrophage cell line RAW264.7 was used. NMRAL1 mRNA expression levels were assessed via RT-qPCR, and inflammatory cytokine levels (TNF-α, IL-1β, IL-17 and IL-23) were measured using ELISA following NMRAL1 siRNA transfection in LPS-treated macrophages.

### Results

We collected colonic mucosa specimens from 20 patients, including groups A,B, C and D with 5 patients in each group. We established a database of DEPs and identified 107 (63 upregulated and 44 downregulated) between group B and group A, 78 (16 upregulated and 62 downregulated) between group D and group C, 45 (8 upregulated and 37 downregulated)

**Data Availability Statement:** Proteomic data are available via ProteomeXchange with identifier PXD032159. Cell experimental data are available on Figshare at the following link: https://doi.org/10.6084/m9.figshare.27950127. Other relevant data

are within the manuscript and its Supporting information files.

**Funding:** This work was supported by Jiangsu Provincial Science and Technology Program Special Funds Project (Key R&D Program for Social Development) (grant number BE2022801 to X.H), Jiangsu Province Higher Education "Qinglan Project" Outstanding Teaching Team Program (Sujiaoshihan [2024] No.2) to X.H, and Jiangsu Province University Superior Discipline Construction Project (Sujiaoyanhan [2019] No.4) to X.H. The funders had no role in study design, data collection and analysis, decision to publish, or preparation of the manuscript.

**Competing interests:** The authors have declared that no competing interests exist.

between group D and group B, and 57 (33 upregulated and 24 downregulated) between group C and group A. Further analysis revealed that the NmrA-like family domain containing 1 (NMRAL1) protein was identified as a DEP specifically associated with group D. The results of in vitro results showed a significant decrease in NMRAL1 mRNA expression in LPS-treated cells ($P<0.001$), which was further reduced in NMRAL1 siRNA-transfected cells ($P<0.0001$), confirming successful transfection. ELISA results revealed that the levels of key inflammatory cytokines (TNF-α, IL-1β, IL-17 and IL-23) were significantly elevated in the LPS-treated model group ($P<0.0001$, $P<0.001$), but these levels were significantly decreased after NMRAL1 siRNA transfection ($P<0.0001$, $P<0.01$, $P<0.05$).

## Conclusion

NMRAL1 is identified as a key differentially expressed protein in AS patients with gut inflammation. Knockdown of NMRAL1 significantly reduced the levels of inflammatory cytokines, suggesting its potential role in the pathogenesis of AS and gut inflammation, and as a possible therapeutic target.

## 1. Introduction

Ankylosing spondylitis (AS) is a chronic and progressive inflammatory disease which mainly invades the fibrous and synovial joints of the sacroiliac and spinal joints [1]. Peripheral involvement includes enthesitis, dactylitis and arthritis. The most common extra-articular manifestations are uveitis, psoriasis and inflammatory bowel disease [2]. The etiology and pathogenesis of AS remain unclear but are thought to be related to factors such as genetics, oxidative stress, mineral metabolism disorders, smoking, infections, and gut microbiota [3]. The gut microbiota is a complex microbial ecosystem that plays a crucial role in the development of various autoimmune diseases, including multiple sclerosis, rheumatoid arthritis, type 1 diabetes and systemic lupus erythematosus [4]. It is reported that up to 70% of AS patients have subclinical intestinal inflammation, and 5–10% suffer from more severe intestinal inflammation, even inflammatory bowel disease (IBD) [5]. In addition, it has been found that gut inflammation is often closely associated with high disease activity of AS and more severe sacroiliac joint bone marrow edema [6]. Our research group's previous clinical observation found that the intervention of intestinal symptoms in patients with AS accompanied by gut inflammation can not only improve abdominal pain and diarrhea, but also better alleviate the pain symptoms, reduce disease activity and even reverse imaging findings. Therefore, we boldly speculate that there is some mysterious correlation between AS and gut inflammation.

Proteomics researchers uses different approaches to look for target proteins, quantify signal proteins, and analyze protein interactions [7–9] in efforts to investigate the nature of diseases. Most proteomics studies on AS have focused on serological markers [10, 11], and no studies have investigated the relationship between AS and gut inflammation at the level of colon mucosal proteomics. In this study, we performed colon mucosal proteomics to investigate the relationship between AS and gut inflammation and to identify any differentially expressed proteins (DEPs) unique to AS combined with gut inflammation. We also investigated the effect of differentially expressed proteins on inflammatory cytokines through in vitro experiments to explore its potential role in the pathogenesis of AS and gut inflammation. Fig 1 shows our experimental workflow.

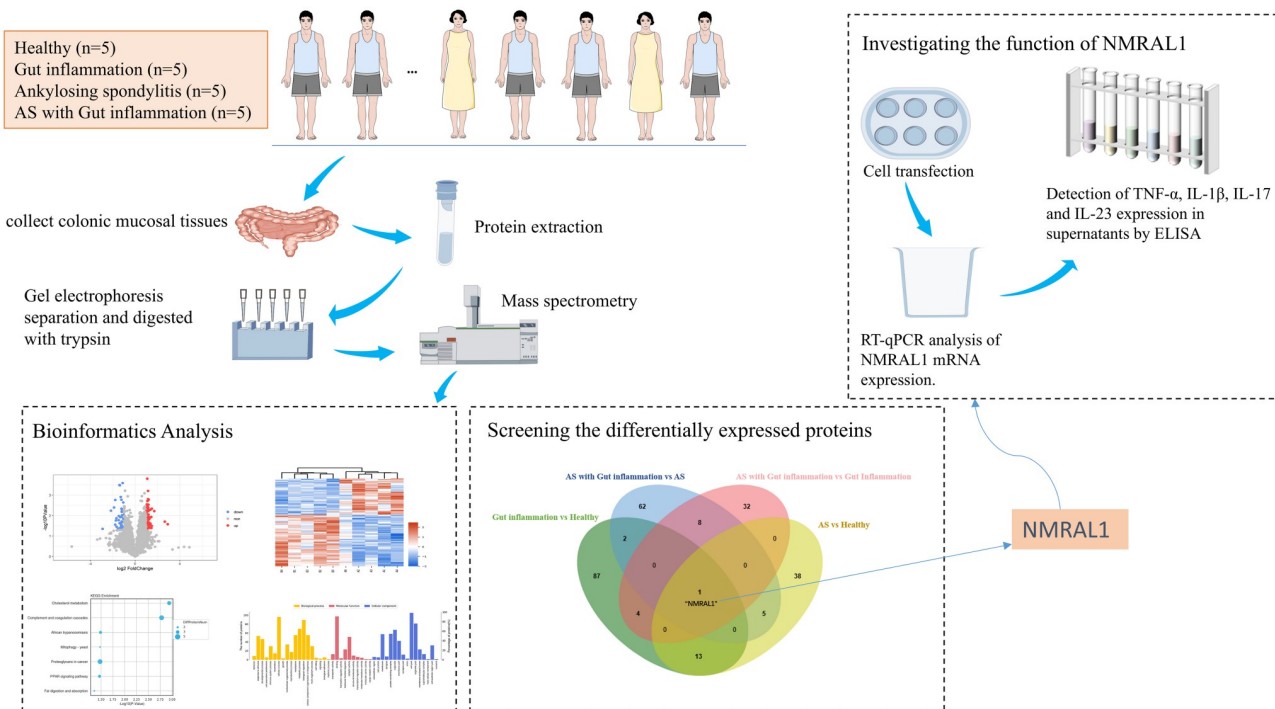

**Fig 1. The experimental workflow of the colonic mucosal proteomics study on AS with intestinal inflammation.** Colon mucosal tissues were collected from four groups, with 5 individuals in each group: healthy individuals, patients with gut inflammation, patients with AS, and patients with AS combined with gut inflammation. These tissues underwent protein lysis, gel electrophoresis separation, trypsin digestion, and subsequent mass spectrometry analysis. The differentially expressed proteins identified were annotated for their biological functions using bioinformatics tools. NMRAL1 was found to be a differentially expressed protein common to all four groups. In vitro experiments were conducted to explore the impact of NMRAL1 on the expression of inflammatory cytokines TNF-α, IL-1β, IL-17, and IL-23, employing cell transfection methods, RT-qPCR, and ELISA assays.

## 2 Materials and instruments

### 2.1 Materials

We collected tissue samples from outpatients and inpatients treated at the Department of Rheumatology and Immunology or the Digestive Endoscopy Center, Jiangsu Provincial Hospital of Traditional Chinese Medicine, between July 2020 and December 2020. AS patients diagnosed according to the modified New York Criteria for Ankylosing Spondylitis (American College of Rheumatology, 1984) were included in the AS group. Patients were included in the gut inflammation group when they underwent colonoscopy and found acute or chronic lesions in the biopsy specimen [12]. Patients who met the inclusion criteria of both AS and gut inflammation were included in the AS combined with gut inflammation group. Individuals without AS and gut inflammation were included in the healthy group. Subjects with the diagnosis of other intestinal diseases (such as inflammatory bowel disease), autoimmune diseases, infectious diseases (such as tuberculosis or cytomegalovirus, etc.), and a history of neurological cognitive disease were excluded from the four groups. In order not to affect the results of the study, patients who have used biological agents within one year are not allowed to join the group. Finally, 10 men aged 23 to 88 (mean: 44) and 10 women aged 31 to 69 (mean: 53.5) were included in this study.

## 2.2 Main instruments

Nano Elute chromatography system and timsTOF Pro mass spectrometer (Bruker, Germany); low-temperature high-speed centrifuge (5430R;Eppendorf, Germany); Agilent 1260 Infinity II HPLC system (Agilent, USA); electrophoresis apparatus (Bio-Rad, USA); ultrasonic disintegrator (JY96-IIN; Ningbo Xinzhi Biological Technology Co. Ltd., China); vacuum centrifugal concentrator (LNG-T98; Jiangsu Taicang Huamei Technology Co. Ltd., China); MP Fastprep-24 homogenizer (Fastprep-24 5G; MP, USA); thermostatic incubator (GNP-9080; Shanghai Jinghong Chemical Co. Ltd., China); electronic balance (AX324Z; OHAUS, USA); compact thermostatic mixer (HCM-100 pro; Beijing Dalong Technology Co. Ltd., China); vortex shaker (GENIE Vortex-2; Scientific Industries, USA); NanoDrop 2000 spectrophotometer and Multiscan FC microplate reader (Thermo Fisher Scientific, USA).

## 3 Methods

### 3.1 Sample collection

A total of 20 samples were collected for colonic mucosa study, including five from the healthy group (group A; A1, A2, A3, A4, A5), five from the gut inflammation group (group B; B1, B2, B3, B4, B5), five from the AS group (group C; C1, C2, C3, C4, C5), and five from the AS combined with gut inflammation group (group D; D1, D2, D3, D4, D5). Colonoscopy was performed to randomly collect a piece of colon mucosal tissue (approximately 200–500 mg) from every included patient. Any nontarget tissues such as connective tissue and adipose tissue were immediately removed. The samples were washed in phosphate-buffered saline to remove blood and debris, put in liquid nitrogen, and then stored at -80˚C within 2 hours. This study was approved by the Ethics Committee of the Affiliated Hospital of Nanjing University of Traditional Chinese Medicine(Ethics No. 2020NL-062-03). All participants signed informed consent forms.

### 3.2 Proteomic analysis of colonic mucosa

**3.2.1 Protein extraction.** A certain amount of SDT lysis solution (4% SDS, 100 mM Tris-HCl, pH 7.6) was added to each sample, followed by homogenization in a Lysing Matrix A tube (24×2, 6.0 M/S, 60 s), boiling in a water bath for 10 minutes, and centrifugation at 14,000$g$ for 15 minutes. The supernatant was filtered with a 0.22-µm centrifuge tube, and the filtrate was collected. The protein concentration was measured with the bicinchoninic acid method, and then the samples were aliquoted and stored at -80˚C.

**3.2.2 Sodium dodecyl sulfate–polyacrylamide gel electrophoresis (SDS-PAGE).** Loading buffer (6×) was added to each protein sample (20 µg), which was boiled in a water bath for 5 minutes and then separated by 12% SDS-PAGE (250 V, 40 minutes) and stained with Coomassie blue.

**3.2.3 Filter-aided sample preparation (FASP).** In a centrifuge tube, a protein sample (80 µg) and the dithiothreitol stock solution were added and mixed to a final concentration of 100 mM, followed by boiling in a water bath for 5 minutes. Once the sample had cooled to room temperature, 200 µL of UA buffer (8M Urea, 150mM Tris-HCl, pH 8.5) was added and mixed, and then the sample was transferred to a 30-kD ultrafiltration centrifuge tube and centrifuged at 12,500$g$ for 15 minutes. The filtrate was discarded, and the above procedure was repeated once. One hundred microliters of IAA buffer (100 mM IAA in UA) was added, and the sample was placed on a shaker at 600 rpm for 1 minute, followed by 30-minute reaction in the dark at room temperature and then centrifugation at 12,500$g$ for 15 minutes. Next, 100 µL of UA buffer was added, and the sample was centrifuged at 12,500$g$ for 15 minutes. The above

procedure was repeated twice. Then 100 μL of 40 mM NH4HCO3 solution was added, and the sample was centrifuged at 12 500 g for 15 minutes. The procedure was repeated twice. Next, 40 μL of trypsin buffer (4 μg of trypsin in 40 μL of 40 mM NH4HCO3 solution) was added, and the sample was placed on a shaker at 600 rpm for 1 minute and then 16–18 hours at 37°C. The sample was transferred to a collection tube and centrifuged at 12,500$g$ for 15 minutes. Twenty microliters of 40 mM $NH_4HCO_3$ solution was added, and the sample was centrifuged at 12,500$g$ for 15 minutes. The filtrate was collected, desalted with a $C_{18}$ cartridge, lyophilized, and suspended in 40 μL of 0.1% formic acid solution, followed by peptide quantification at 280 nm.

**3.2.4 Mass spectrometry (MS).** The mixed peptides were separated with the NanoElute system with a nanoliter flow rate (solution A: 0.1% formic acid aqueous solution; solution B: 0.1% formic acid acetonitrile aqueous solution [100% acetonitrile]). The column was balanced with 100% solution A, and the samples were loaded with an autosampler to the column (IonOpticks, Australia, 25 cm × 75 μm, C18 column, 1.6 μm) for separation over a 2-hour gradient at 300 nL/min. The gradient included 3% solution B for minutes 0–5; 3–28% solution B for minutes 5–95; 28–38% solution B for minutes 95–110; 38–100% solution B for minutes 110–115; and 100% solution B for minutes 115–120.

The PASEF mode of the timsTOF Pro mass spectrometer (Bruker, Germany) was used for MS analysis. The parameters were as follows: analysis time: 120 minutes; detection method: positive ion; precursor ion scan range: 100–1700 $m/z$; ion mobility 1/K0: 0.6–1.6 V·s/cm$^2$; ion accumulation or release time: 100 ms; ion utilization rate: 100%; capillary voltage: 1500 V; drying gas rate: 3 L/min; and drying temperature: 180°C. The settings of PASEF were 10 MS/MS scans (total cycle time: 1.16 s), charge range 0–5, dynamic rejection time 0.4 minutes, ion target intensity 20,000, ion intensity threshold 2500, and collision-induced dissociation fragmentation energy 42 eV.

## 3.3 Cellular experiments to investigate the effects of NMRAL1 on inflammatory factors

**3.3.1 Cell culture.** RAW264.7 macrophages were obtained from the Cell Bank of the Chinese Academy of Sciences. The cells were cultured in DMEM medium supplemented with 10% fetal bovine serum (FBS), 50μM β-mercaptoethanol, 20 mM HEPES, 10 mM sodium pyruvate, 100μg/ml streptomycin, and 100 U/ml penicillin. The cells were maintained in a 5% CO2 incubator.

**3.3.2 NMRAL1 siRNA transfection in RAW264.7 macrophages.** P3 generation cells were plated in 6-well plates, with 1×10^4 cells seeded in 2 mL of medium per well. The cells were divided into three groups: blank control group (NC), model group (Mod), and model + transfection group (Mod+Trans). On the day of transfection, the cells reached 50–60% confluence. For the model +transfection group, EndofectinTM RNAi transfection reagent (Gene-Copoeia, USA) and NMRAL1 siRNA (Cas9X, Suzhou, China) were diluted in optiMEM (Gibco, USA). A total of 250 μL of the transfection mixture was gently added dropwise to the RAW264.7 macrophages, and the cells were incubated at 37°C in a 5% CO2 incubator for 24 hours. After 24 hours of transfection, 1μg/mL LPS (Sigma, USA) was added to both the model group and the model + transfection group, and the cells were incubated at 37°C in a 5% CO2 incubator for an additional 12 hours. After this incubation, the supernatants and RNA were collected for further analysis.

Although RAW264.7 macrophages produce TNF-α, IL-1β, and IL-23 in response to LPS stimulation, they secrete relatively low levels of IL-17. Therefore, to enhance the detection of IL-17, a co-culture system was employed. After LPS stimulation, the supernatant from the RAW264.7 macrophages was collected and co-cultured with Jurkat human T lymphocyte

leukemia cells (iCell Bioscience Inc, Shanghai, China). Following incubation, the supernatant from the co-culture was collected and used for subsequent ELISA analysis to detect IL-17 expression levels.

**3.3.3 Quantitative Real-Time PCR (RT-qPCR) analysis of NMRAL1 mRNA expression.** Total RNA was isolated using an RNA extraction kit (Accurate Biology, Nanjing, China), and reverse transcription was performed using a reverse transcription kit (Vazyme, Nanjing, China). The amplification was conducted in a total volume of 20μL (Vazyme, Nanjing, China) containing 0.4μL (10μM) forward and reverse primers for NMRAL1, 10μL of 2× ChamQ Blue Universal SYBR, 1μL of reverse-transcribed cDNA, and 8.2μL of ddH2O. The PCR conditions were as follows: initial denaturation at 95˚C for 30 seconds, followed by 40 cycles of denaturation at 95˚C for 5 seconds, and annealing/extension at 60˚C for 30 seconds. The relative expression levels of NMRAL1 mRNA were normalized to β-actin and calculated using the $2^{-\Delta\Delta Ct}$ method. The primers used in the present study are listed as follows (5'-3'). NMRAL1 FORWARD: `TGC GGC TGC CTT GCT ATT T`; NMRAL1 REVERSE: `TCA GGA GTT GTC TTG GCA TGA`; Actb FORWARD: `GGC TGT ATT CCC CTC CAT CG`; Actb REVERSE: `CCA GTT GGT AAC AAT GCC ATG T`.

**3.3.4 ELISA for detection of inflammatory cytokines in supernatants.** Supernatants from RAW264.7 macrophage cultures were collected to measure the levels of TNF-α, IL-1β, and IL-23 using commercial ELISA kits (Jinyibai, Nanjing, China). After co-culture, supernatants were harvested to detect IL-17 levels by ELISA. All measurements were performed according to the manufacturer's instructions, and each sample was analyzed in triplicate.

## 3.4 Data analysis

**3.4.1 MS file processing.** MaxQuant was used for quantitative calculation of label-free quantitative proteomic data.

**3.4.2 Database selection.** The database Uniprot_HomoSapiens_20367_20200226 (http://www.UniProt.org) was used to identify proteins.

**3.4.3 Parameters for qualitative and quantitative protein analysis.** The label-free quantitation algorithm was used for quantitative analysis, and MaxQuant was used for qualitative matching. The parameters of the database are as follows: Maximum missed cleavages are 2. Fixed modifications are carbamidomethyl(C). Variable modifications are oxidation(m), acetyl (protein N-term). Database is Uniprot_HomoSapiens_20367_20200226 and database pattern is target-reverse true. Include contaminants are target-reverse.

A global FDR control strategy was applied to adjust p-values for multiple testing and the cutoff of global false discovery rate (FDR) for peptide and protein identification is set to less than or equal to 0.01. This approach ensures that the likelihood of type I errors due to multiple comparisons is adequately controlled, maintaining the integrity of our findings.

## 3.5 Statistical analyses

Graphpad7.0 was used for statistical analysis. Quantitative variables were presented as mean ± standard deviation (SD) and the data between multiple groups were analyzed using one-way ANOVA. A *P* value < 0.05 was accepted as significant.

## 4 Results

### 4.1 Basic characteristics of the participants

A total of 5 healthy individuals, 5 gut inflammation patients, 5 AS patients and 5 AS with gut inflammation patients were enrolled into the colon mucosal tissue study. The CRP level was

**Table 1. Basic characteristics of the participants with intestinal mucosal specimens collected.**

| Characteristics | AS (n = 5) | gut inflammation(n = 5) | AS with gut inflammation(n = 5) | Healthy individuals(n = 5) |
|---|---|---|---|---|
| Age, years | 47.2±17.06 | 49.6±20.60 | 48.8±11.82 | 49.4±12.64 |
| Gender(male/female) | 4/1 | 2/3 | 3/2 | 1/4 |
| CRP, mg/L | 16.28±6.72 | 6.62±4.08 | 18.63±7.84 | - |

(6.62±4.08) mg/L in gut inflammation group. The CRP level was (16.28±6.72) mg/L in AS group. And the CRP level was (18.63±7.84) mg/L in AS with gut inflammation group (Table 1).

## 4.2 Qualitative and quantitative analysis

A total of 7039 proteins and 81141 peptides were identified in the database. S1 Table provides detailed information on the proteins identified in our study, including Protein ID, Gene Name, Number of Peptides, and Molecular Weight. S2 Table provides detailed information on the peptides identified in our study, encompassing Sequence, Length, Mass, and Score.

For the Quantitative results, we screened duplicate data to select data with at least 50% non-empty values. A DEP was defined as one that was up or downregulated by at least 2-fold ($P<0.05$) between two groups. The number of significant DEPs is shown in Table 2. Principal Component Analysis (PCA) revealed clear distinctions between the different study groups, demonstrating that the proteomic profiles of colonic mucosal tissues vary significantly among patients with AS, gut inflammation, AS combined with gut inflammation and healthy individuals (S1 Fig). S3 Table summarizes the proteins identified as upregulated or downregulated, with upregulation marked in yellow and downregulation in green. The table also lists the fold change values for these differentially expressed proteins in each group.

## 4.3 Bioinformatics analysis

**4.3.1 Volcano plot.** We used the fold change in protein expression and the p-value between the two groups to generate a volcano plot, highlighting the significant differences in protein expression between the sample groups. DEPs were identified between groups B and A, groups D and C, groups D and B, and groups C and A (Fig 2a–2d).

Among the highly differentially expressed proteins identified, SPATS2L, PCSK1, and FAM98A showed significant upregulation in group B compared to group A. COL5A1 was notably upregulated in group D compared to group C, while APCS and ALPL were significantly different between groups D and B. Additionally, CARHSP1 was identified as highly differentially expressed between groups C and A. These proteins are marked in the volcano plots (Fig 2a–2d).

**Table 2. Quantitative analysis.**

| Group comparison | Upregulated DEPs | Downregulated DEPs | Total |
|---|---|---|---|
| B/A | 63 | 44 | 107 |
| C/A | 33 | 24 | 57 |
| D/A | 59 | 73 | 132 |
| C/B | 44 | 9 | 53 |
| D/B | 8 | 37 | 45 |
| D/C | 16 | 62 | 78 |

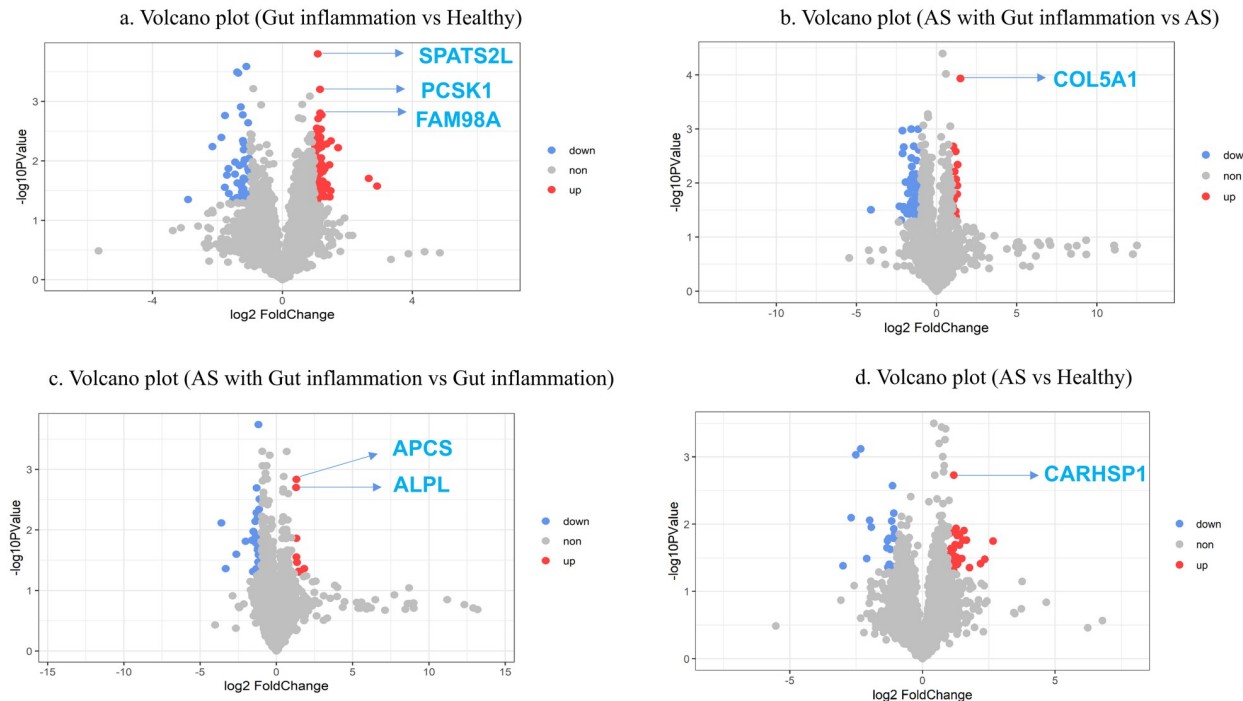

**Fig 2. Volcano plots of differentially expressed proteins between groups.** The x-coordinate is the multiple of the difference in expression level (log2), and the y-coordinate is the P value (log10). Each dot represents a protein; red and blue dots represent upregulated and downregulated proteins, respectively; and gray dots represent non-differentially expressed proteins.

**4.3.2 Cluster analysis.** Cluster analysis revealed that the differentially expressed proteins (DEPs) exhibited high intergroup heterogeneity, indicating low similarity in protein expression profiles between the different groups (B/A, D/C, D/B, and C/A). Conversely, there was high intragroup homogeneity, meaning that the protein expression profiles within each group were highly similar, thereby reinforcing the distinctiveness of the DEPs associated with each group comparison (Fig 3a–3d).

**4.3.3 Gene Ontology (GO) annotation and enrichment analysis.** GO is a standardized functional classification system, and its top-level annotations include biological processes, molecular functions, and cellular components. We compared the GO annotations for the B/A, D/C, D/B, and C/A DEPs (Fig 4a–4d).

The InterPro database was used to analyze the enrichment of the functional domains of the DEPs. GO enrichment analysis showed that the DEPs of group B versus group A mainly had the biological functions of phosphatidylcholine-sterol *O*-acyltransferase activator activity, positive regulation of cholesterol esterification, platelet alpha granule lumen, lipoprotein metabolic process, negative regulation of cytokine secretion involved in immune response, and embryonic skeletal system development.

GO enrichment analysis showed that the DEPs of group D versus group C mainly had biological functions related to very-low-density lipoprotein particles, chloride channel activity, lipoprotein metabolic processes, histone demethylase activity (H3-K4 specific), apoptotic DNA fragmentation, and cellular oxidant detoxification.

In the group D versus group B comparison, the main biological functions of the DEPs were zinc ion binding, myoblast differentiation, transcription regulatory region DNA binding, cell growth, epithelial-to-mesenchymal transition, regulation of MDA-5 signaling pathway, and regulation of RIG-I signaling pathway.

a. Cluster analysis (Gut inflammation vs Healthy)

b. Cluster analysis (AS with Gut inflammation vs AS)

c. Cluster analysis (AS with Gut inflammation vs Gut inflammation)

d. Cluster analysis (AS vs Healthy)

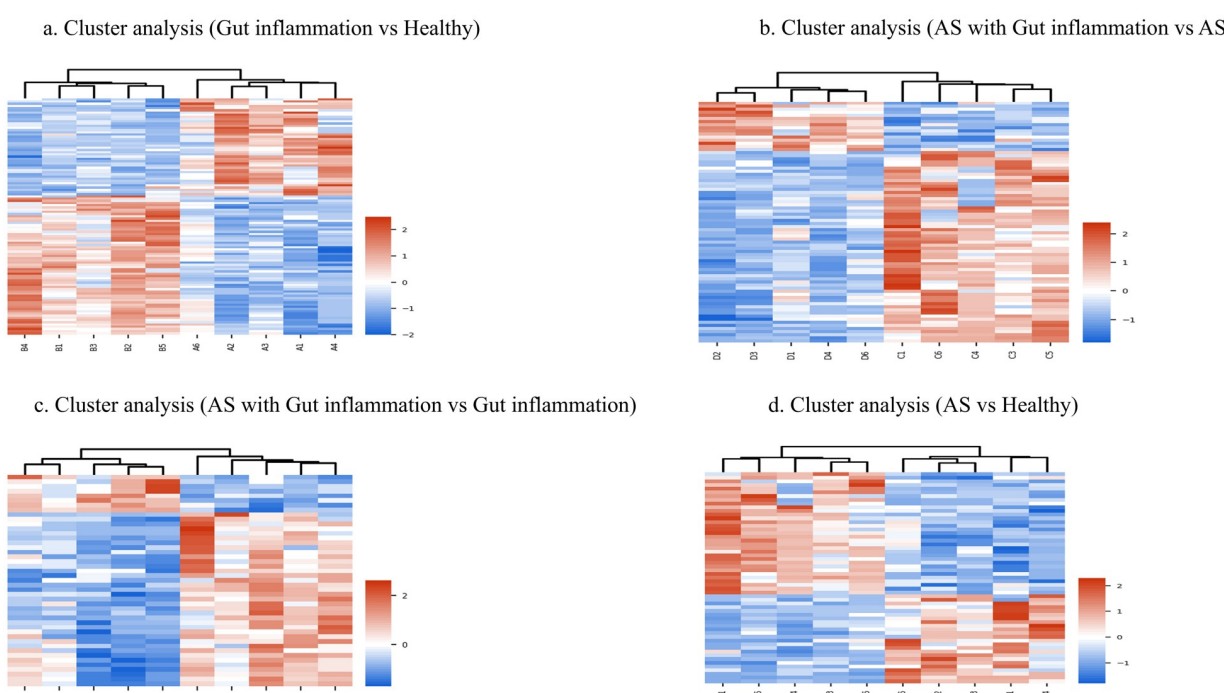

**Fig 3. Heat map of differentially expressed proteins between groups.** The heat map shows differentially expressed proteins between groups, with highly expressed proteins in red and lowly expressed proteins in blue.

a. Histogram of GO annotations (Gut inflammation vs Healthy)

b. Histogram of GO annotations (AS with Gut inflammation vs AS)

c. Histogram of GO annotations (AS with Gut inflammation vs Gut inflammation)

d. Histogram of GO annotations (AS vs Healthy)

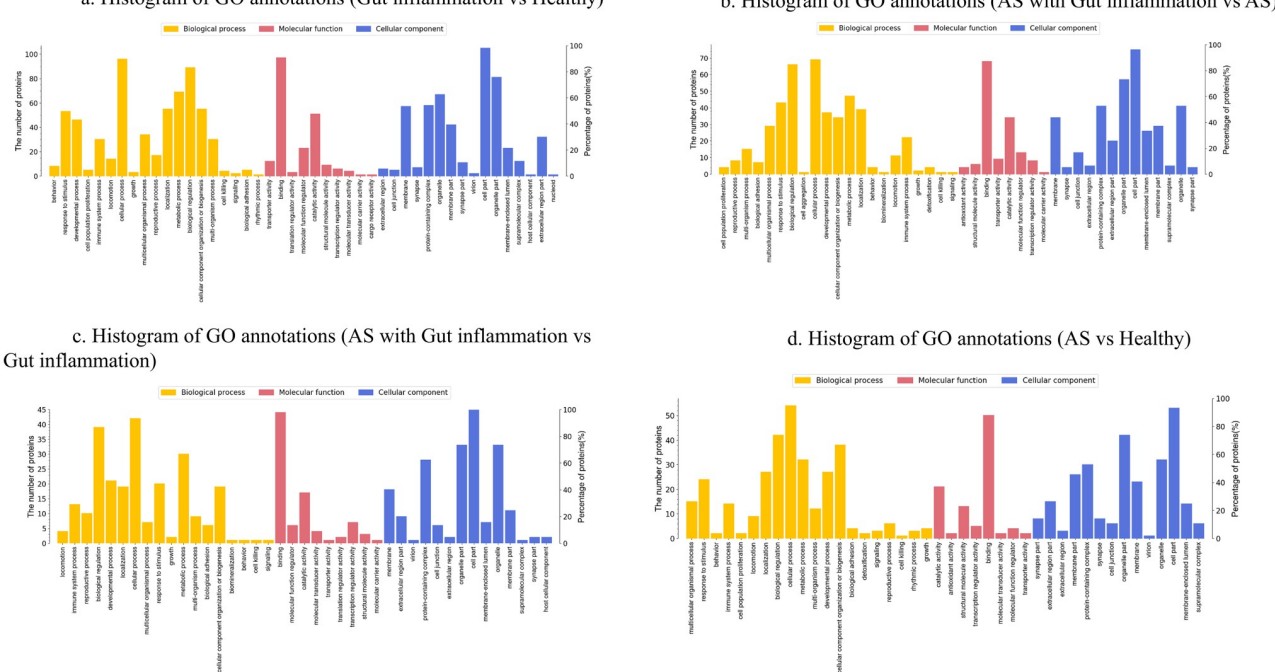

**Fig 4. GO annotations of differentially expressed proteins between groups.** This figure displays the GO annotations of differentially expressed proteins between groups, categorized into biological processes, molecular functions, and cellular components.

Between group C and group A, the DEPs had the main biological functions of skeletal muscle fiber adaptation, smooth muscle contractile fiber, mitochondrial respiratory chain complex I assembly, vascular smooth muscle contraction, mitochondrial outer membrane permeabilization involved in programmed cell death, L-phenylalanine metabolic processes, sarcomere organization, negative regulation of B cell proliferation, and negative regulation of interleukin-5 production.

**4.3.4 Kyoto Encyclopedia of Genes and Genomes (KEGG) analysis.** The DEPs were analyzed for their enrichment in KEGG pathways. KEGG analysis showed that the DEPs of group B versus group A were enriched in the pathways of cholesterol metabolism, complement and coagulation cascades, proteoglycans in cancer, PPAR signaling pathway, and fat digestion and absorption. The DEPs of group D versus group C were enriched in PPAR signaling, arachidonic acid metabolism, cholesterol metabolism, and proximal tubule bicarbonate reclamation. The DEPs of group D versus group B were enriched in RNA transport. The DEPs of group C versus group A were enriched in thermogenesis, retrograde endocannabinoid signaling, apoptosis-fly, and asthma (Fig 5a–5d).

**4.3.5 Screening of DEPs.** The DEPs of group B (gut inflammation) versus group A (healthy individuals) and of group D (AS combined with gut inflammation) versus group C (AS) were aligned to identify their common DEPs. These were NMRAL1, ORM1, and APOA2. The only DEP shared in common by the group D (AS combined with gut inflammation) versus group B (gut inflammation) comparison and the group C (AS) versus group A (healthy individuals) comparison was NMRAL1). Finally, aligning these two sets with each other revealed that NMRAL1 was the DEP that identified group D. The Venn diagram below vividly demonstrates the process of screening DEPs (Fig 6).

a. KEGG analysis (Gut inflammation vs Healthy)

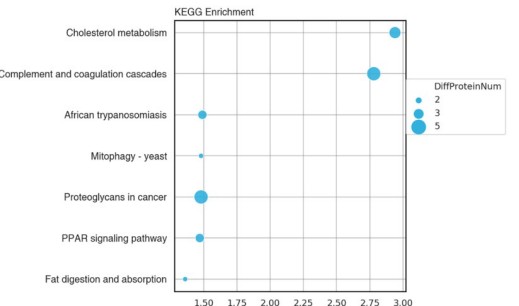

b. KEGG analysis (AS with Gut inflammation vs AS)

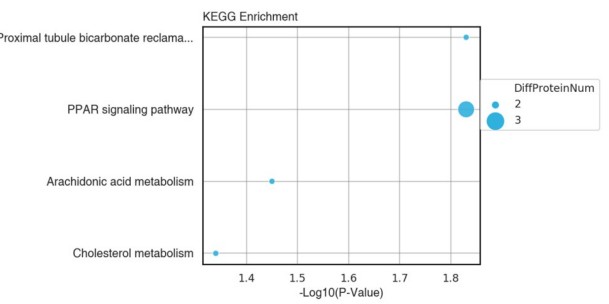

c. KEGG analysis (AS with Gut inflammation vs Gut inflammation)

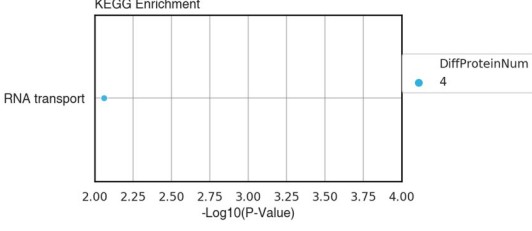

d. KEGG analysis (AS vs Healthy)

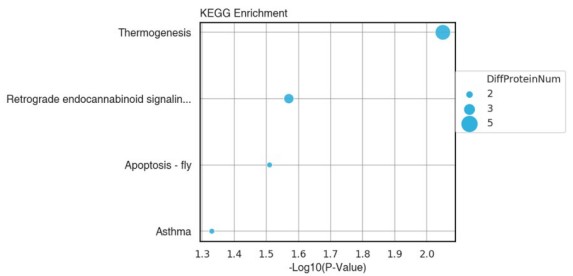

**Fig 5. KEGG pathway enrichment bubble plot of differentially expressed proteins between groups.** The bubble plot represents KEGG pathway enrichment analysis of differentially expressed proteins between groups. Each bubble corresponds to a pathway, with bubble size indicating the number of associated DEPs and color reflecting the significance level (p-value).

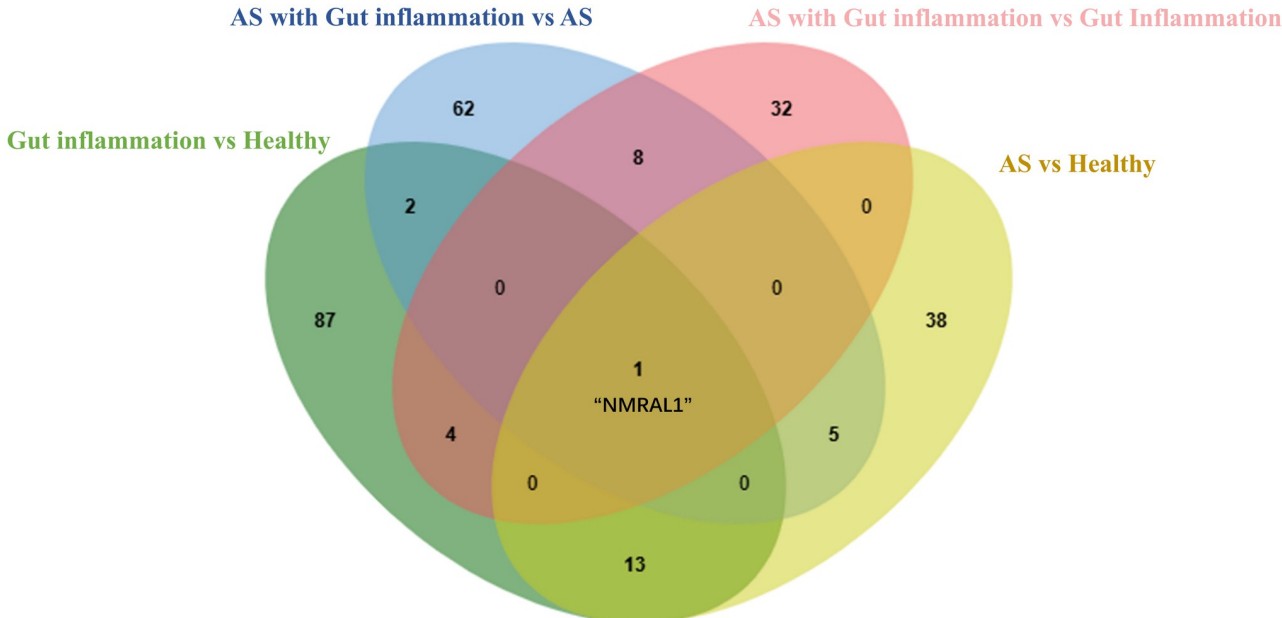

**Fig 6. Venn diagram illustrating common and unique DEPs across group comparisons.** Venn Diagram showing NMRAL1 as the unique DEP identifying Group D (AS combined with gut inflammation) across various group comparison.

## 4.4 Changes in inflammatory cytokine expression in NMRAL1 siRNA-transfected macrophages

To explore the inflammation-related function of NMRAL1 protein, we utilized the murine monocyte/macrophage cell line RAW264.7, an established model for inflammation studies. RT-qPCR results showed that, compared to the blank control group, the relative expression of NMRAL1 mRNA was significantly decreased in the model group treated with LPS ($P<0.001$). The reduction in NMRAL1 mRNA expression was even more pronounced in the NMRAL1 siRNA-transfected macrophages ($P<0.0001$), indicating successful transfection of NMRAL1 siRNA (Fig 7).

ELISA results demonstrated that the expression levels of TNF-α, IL-1β and IL-23 were significantly increased in the model group compared to the blank control group ($P < 0.0001$, $P<0.001$). Notably, IL-17 levels were measured following co-culture of LPS-stimulated RAW264.7 macrophage supernatants with Jurkat cells to enhance IL-17 detection. After NMRAL1 siRNA transfection, the expression levels of all these inflammatory cytokines, including IL-17, were significantly decreased in the model + transfection group ($P<0.0001$, $P<0.01$, $P<0.05$) (Fig 8a–8d).

## 5 Discussion

A growing body of evidence suggests a close relationship between intestinal inflammation and ankylosing spondylitis [13–15]. Campos JF etal. recently discovered that patients with spondyloarthritis (SpA) have a high prevalence of microscopic gut inflammation, and fecal calprotectin levels can effectively identify microscopic inflammation, serving as a biomarker for microscopic intestinal inflammation in SpA patients [16]. Based on the similar pathogenesis between inflammatory bowel disease and SpA, some scholars proposed the intestinal joint axis theory [17]. Previous studies on AS biomarkers have primarily focused on serum proteomics,

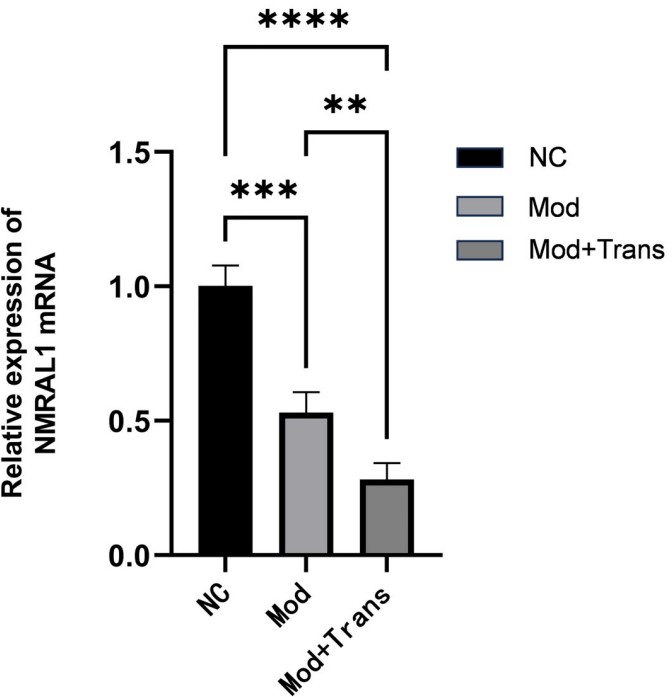

**Fig 7. Effect of NMRAL1 siRNA transfection on NMRAL1 mRNA expression in RAW264.7 macrophages.** The relative expression of NMRAL1 mRNA was analyzed using RT-qPCR in three groups: blank control group (NC), model group treated with LPS (Mod), and model + transfection group (Mod + Trans). Data are presented as means ± SE of three repeats. ****$P < 0.0001$, ***$P < 0.001$, **$P < 0.01$.

lipidomics, and metabolomics [18–20]. The present study is the first to investigate the relationship between AS and gut inflammation through colon mucosal proteomics.

In this study, we collected colon mucosal samples to compare the DEPs between AS and gut inflammation. A total of 20 patients were included in this study, with five patients each in the healthy group (group A), the gut inflammation group (group B), the AS group (group C), and the AS combined with gut inflammation group (group D). We identified a total of 81,141 peptides and 7039 proteins. Further analysis demonstrated abnormal protein expression in colon mucosa in groups B, C, and D relative to group A. We identified one DEP that distinguished group D from the other groups, suggesting that the DEPs identified in this study represent intergroup differences. By GO and KEGG analysis, we identified NMRAL1 as the DEP identifying AS combined with gut inflammation.

NMRAL1, also called NmrA-like family domain-containing protein 1 or HSCARG, is an nicotinamide adenine dinucleotide phosphate (NADPH) sensor protein [21]. The crystal structure of NMRAL1 consists of an N-terminal NmrA domain containing a Rossmann fold and a smaller C-terminal domain [22]. NADPH is an allosteric regulator of the structure and function of NMRAL1. Under normal redox status, NMRAL1 binds to coenzyme NADPH via its Rossmann fold, which stabilizes NMRAL1 as an asymmetric dimer in the cytoplasm. Decrease in the NADPH/NADP+ ratio results in NMRAL1 existing mainly in its NADPH-free monomeric form [22, 23]. Previous studies have shown that NMRAL1 participates in a variety of biological processes, including redox homeostasis [24, 25], innate immunity [26, 27], cellular antiviral response [28] and DNA damage response [29]. For example, NMRAL1 could represses the cellular ROS generation by inhibiting mRNA and protein expression of p47phox

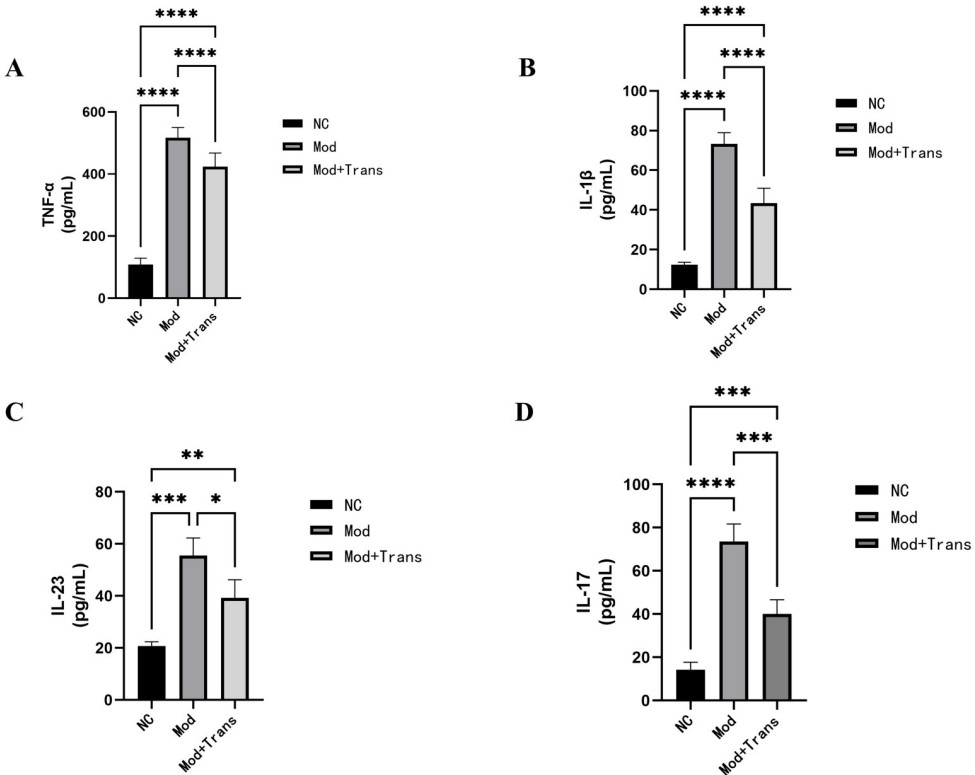

**Fig 8. Effects of NMRAL1 siRNA transfection on the expression of inflammatory cytokines in RAW264.7 macrophages.** ELISA analysis of TNF-α (A), IL-1β (B), IL-23 (C), and IL-17 (D) expression levels in RAW264.7 macrophages. Cells were divided into the following groups: blank control group (NC), model group treated with LPS (Mod), and model + transfection group (Mod + Trans). Compared to the blank control group, the model group exhibited a significant increase in the levels of all measured cytokines ($P<0.0001$, $P<0.001$). In contrast, NMRAL1 siRNA transfection significantly reduced the expression of these cytokines in the Mod + Trans group ($P<0.0001$, $P<0.001$, $P<0.05$). Data are presented as means ± SE of three repeats. ****$P < 0.0001$, ***$P < 0.001$, **$P < 0.01$, *$P < 0.05$.

(a subunit of NADPH oxidase), suggesting that NMRAL1 is a novel regulator regulating NADPH oxidase activity and ROS balance [24]. In addition, NMRAL1 has a close biological relationship with cancer. Zang W et al. [30] recently demonstrated the up-regulation of HSCARG exacerbates mammary tumorigenesis. They found that compared with wild type or heterozygous PyMT mice, PyMT transgenic mice with NMRAL1 gene knockout show delayed mammary tumor occurrence. Wang J et al. [31] identified NMRAL1 as a potential novel risk gene for schizophrenia, regulated by the rs2270363 variant. This gene may confer susceptibility to schizophrenia by influencing neural development and modulating dendritic spine density.

Inflammation serves as a critical pathogenic mechanism in AS, typically characterized by the dysregulated production of various inflammatory cytokines. Among these, TNF-α has been identified as potential serum biomarkers for AS [32]. TNF-α is widely recognized as a central inflammatory cytokine in AS. Researchers have found that high concentrations of TNF-α can induce METTL14-mediated m6A modification of the ELMO1 3′ UTR, enhancing the migration of AS mesenchymal stem cells, thereby promoting the progression of AS through pathological osteogenesis [33]. Liu B et al. [34] recently found that anti-TNF-α therapy attenuate arthritis progression and alter the gut microbiota in proteoglycan-induced ankylosing spondylitis in mice. A mendelian randomization study found that IL-1β, a pro-

inflammatory cytokine, plays a key role in the pathogenesis of AS [35]. Recent studies have shown that AS patients present with significantly higher mean plasma levels of IL-1β when compared to healthy controls. Moreover, the IL-1β rs2853550 AG genotype has been identified as a genetic variant that heightens the risk of developing AS within the Chinese population [36]. The IL-23/IL-17 axis plays an important role in both AS and gut inflammation. This axis mainly includes IL-23, the IL-23 receptor (IL-23R), signal transducer and activator of transcription 3 (STAT3), Janus kinase 2 (JAK2), and IL-17, among others [37]. Several studies have found that serum levels of IL-17 and IL-23 are significantly elevated in AS patients, and the increase in IL-23 levels is more pronounced in patients with active disease [6, 38, 39]. Qaiyum et al. [40] observed that in the context of subclinical intestinal inflammation, activated Paneth cells are prone to producing IL-23, which further stimulates the production of IL-17 by immune cells such as Th17 cells, type 3 innate lymphoid cells (ILC3), and mucosa-associated invariant T (MAIT) cells, which are pivotal in the pathogenesis of spondyloarthritis. Our experimental results demonstrate that NMRAL1 protein plays a significant upstream regulatory role in the expression of key inflammatory cytokines associated with AS. Specifically, the downregulation of NMRAL1, either through LPS treatment or siRNA transfection, led to a notable increase in the levels of TNF-α, IL-1β, IL-17, and IL-23. Conversely, knocking down NMRAL1 resulted in a significant reduction in these cytokines, highlighting its potential as a critical modulator in AS-related inflammation and suggesting its value as a potential therapeutic target in managing the disease.

Our study has certain limitations. Firstly, the number of cases included in our research is relatively small. A larger sample size would enhance the generalizability of our findings and reduce the potential impact of random variation. Future studies with larger sample sizes are needed to validate these results. Secondly, disease activity and duration may also have a relationship with the expression of NMRAL1 protein. Further analyzing the relationship between the NMRAL1 protein and the activity of AS and gut inflammation is needed. Additionally, we did not include serum samples from patients with other inflammatory or autoimmune diseases, which would have further validated the specificity of NMRAL1 as a biomarker for AS combined with gut inflammation.

In conclusion, we used proteomic technology to identify DEPs in the colonic mucosa of patients with AS only, patients with gut inflammation only, patients with AS combined with gut inflammation, and healthy individuals. Among these, we identified NMRAL1 as a specific protein present in the intestinal mucosa of patients with AS complicated by gut inflammation, suggesting it could serve as a potential marker for this condition. Our functional studies using a murine macrophage cell model further demonstrated that NMRAL1 plays a role in regulating key inflammatory cytokines, highlighting its potential significance in the pathogenesis of AS. Moving forward, we plan to expand our sample size and conduct more in-depth investigations into the functional role of NMRAL1, with the goal of providing novel insights and potential therapeutic strategies for the diagnosis and treatment of AS.

## Supporting information

**S1 Fig. Principal component analysis.** This Principal Component Analysis (PCA) plot represents the proteomic profiles of colonic mucosal tissues across four groups: healthy individuals (Group A), patients with gut inflammation only (Group B), patients with AS only (Group C), and patients with AS combined with gut inflammation (Group D). PC1 and PC2 explain 0.28 and 0.12 of the total variance, respectively. The clear separation between the groups along the principal components indicates significant differences in proteomic profiles.
(PDF)

**S1 Table. Protein identification results.** This table provides detailed information on the proteins identified in the study.
(XLSX)

**S2 Table. Peptides identification results.** This table provides detailed information on the peptides identified in the study.
(XLSX)

**S3 Table. Protein differential analysis list.** This Table summarizes the proteins identified as upregulated or downregulated, with upregulation marked in yellow and downregulation in green. The table also lists the fold change values for these differentially expressed proteins in each group.
(XLSX)

## Author Contributions

**Conceptualization:** Miao Cheng, Siqi Xiao, Yujie Guan, Hua Chen, Lei Wang, Xiaojin He.

**Data curation:** Miao Cheng, Siqi Xiao, Shaer Kayi, Yujie Guan, Yingxin Liu, Jianmei Chen, Lei Wang.

**Formal analysis:** Miao Cheng, Siqi Xiao, Yujie Guan, Yingxin Liu, Jianmei Chen, Hua Chen.

**Funding acquisition:** Xiaojin He.

**Investigation:** Siqi Xiao, Shaer Kayi, Jianmei Chen, Hua Chen.

**Methodology:** Miao Cheng, Shaer Kayi.

**Project administration:** Lei Wang, Xiaojin He.

**Resources:** Xiaojin He.

**Supervision:** Hua Chen, Lei Wang, Xiaojin He.

**Validation:** Siqi Xiao.

**Writing – original draft:** Miao Cheng, Shaer Kayi.

**Writing – review & editing:** Miao Cheng.

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
