## [Decision Letter · Decision Letter 0]

5 Nov 2024

PONE-D-24-44456Colon Mucosal Proteomics of Ankylosing Spondylitis versus Gut inflammationPLOS ONE

Dear Dr. He,

Thank you for submitting your manuscript to PLOS ONE.. Therefore, we invite you to submit a revised version of the manuscript that addresses the points raised during the review process.

We look forward to receiving your revised manuscript.

Kind regards,

Sayed Haidar Abbas Raza

Academic Editor

PLOS ONE

Journal Requirements:

2. Please ensure that you have specified a) Did participants provide their written or verbal informed consent to participate in this study?

3. In consent please state in Ethics Method section and manuscript if it is written or verbal. If consent was verbal, please explain a) why written consent was not obtained, b) how you documented participant consent, and c) whether the ethics committees/IRB approved this consent procedure.

Reviewers' comments:

Reviewer's Responses to Questions

**Comments to the Author**

1. Is the manuscript technically sound, and do the data support the conclusions?

Reviewer #1: Yes

Reviewer #2: Yes

2. Has the statistical analysis been performed appropriately and rigorously? 

Reviewer #1: Yes

Reviewer #2: Yes

3. Have the authors made all data underlying the findings in their manuscript fully available?

Reviewer #1: Yes

Reviewer #2: Yes

4. Is the manuscript presented in an intelligible fashion and written in standard English?

Reviewer #1: Yes

Reviewer #2: Yes

5. Review Comments to the Author

Reviewer #1: The manuscript "Colon Mucosal Proteomics of Ankylosing Spondylitis versus Gut inflammation" uses proteomic techniques to identify the differentially expressed proteins in the colon tissues of patients with AS and patients with gut inflammation. The authors collected 20 samples from 4 diff groups: healthy individuals (group A), patients with gut inflammation only (group B), patients with AS only (group C), and patients with AS combined with gut inflammation (group D).The authors found differential protein expression between all the groups. Further analysis revealed that the NmrA-like family domain containing 1 (NMRAL1) protein was identified as a DEP specifically associated with group D. The authors performed a knockdown of NMRAL1 using siRNA and checked for the levels of key inflammatory cytokines and found that the levels were significantly decreased. The manuscript is well written and the conclusions are based on results but I have a few concerns

1. The figure quality is really bad and I cannot read anything even after zooming in. Though the authors have provided tiff figures, the figure quality in the manuscript need to be improved.

2. In the figures the authors have used A, B, C, D to demarcate the diff groups. As a reader it is diff to go back and check what each group is. I suggest that the authors use actual group headings instead of A, B, C, D. For example Healthy, Gut Inflammation, AS and AS with Gut inflammation.

3. In the volcano plots, if the authors can mark the names of some of the highly differentially expressed proteins it will make the reading more interesting.

4. Please make sure to add all the recent references.

5. A graphical abstract can be useful for the reader.

Reviewer #2: Comments:

The manuscript, Colon Mucosal Proteomics of Ankylosing Spondylitis versus Gut Inflammation, examined the impact of NMRAL1 protein on inflammatory cytokines and identified the differentially expressed proteins (DEPs) in the colon tissues of patients with AS and those with gut inflammation and has established this correlation effectively.

It is a well-researched, experimentally supported, and well-presented paper. The study has an excellent design and a comprehensive literature review.

The lowered NMRAL1 expression in response to LPS-induced inflammation, as well as the lower expression with siRNA transfection, suggests that NMRAL1 may play a role in inflammation regulation. The combined evidence from AS samples and in vitro investigations supports the link between NMRAL1 and inflammation-related functions, presumably specific to AS and gut inflammation.

Query 1: However, there is a query regarding collecting colonic mucosa specimens from 20 patients only. This sample size is assumed to be too low for conducting research and generalization of results. Since small sample sizes are more susceptible to random variation and may not represent the broader AS population accurately and there are many factors like age, medication, disease duration, and lifestyle which can influence inflammation in both the gut and joints, thus how the statistical data of your results shows confidence.

Suggestion: In Supplementary Table S1, Title: Protein Identification Results, False Discovery Rate (FDR) and Protein Score can also be added for establishing the confidence in the accuracy of protein identification results.

Except for the queries in the above two points, the overall manuscript presents well-supported conclusions, robust data, statistical analysis, supporting experimental evidence, and clear, concise language and formatting.

6. PLOS authors have the option to publish the peer review history of their article (what does this mean?). If published, this will include your full peer review and any attached files.

Reviewer #1: No

Reviewer #2: **Yes: **Harpreet Kaur

---

## [Author Response · Author response to Decision Letter 0]

16 Nov 2024

Dear Editor and Reviewers,

We sincerely appreciate the time and effort you and the reviewers have dedicated to evaluating our paper. On behalf of all the authors, we would like to express our heartfelt gratitude for your thorough review and the valuable insights and constructive feedback you provided. Your comments have played a crucial role in enhancing the clarity and academic rigor of our work.

Upon receiving your feedback, we promptly convened as a team to thoroughly discuss and carefully address each suggestion. We have revised and refined the manuscript accordingly, ensuring that all adjustments align closely with the reviewers' recommendations. Below, we provide a detailed response to each of the comments raised.

#Journal Requirements:

Response: 

Thank you for your guidance. We have carefully revised the manuscript to fully comply with PLOS ONE's style requirements, including file naming conventions, as specified in the provided templates.

2. Please ensure that you have specified a) Did participants provide their written or verbal informed consent to participate in this study?

Response: 

Thank you for your attention to these details. We confirm that all participants provided written informed consent, as indicated in the manuscript (Line 127).

3. In consent please state in Ethics Method section and manuscript if it is written or verbal. If consent was verbal, please explain a) why written consent was not obtained, b) how you documented participant consent, and c) whether the ethics committees/IRB approved this consent procedure.

Response: 

We have indicated in the Methods section (Line 127) that all participants signed informed consent forms.

 Response: 

We apologize for the oversight regarding the grant information. Upon resubmission, we will ensure that the correct grant numbers are provided in the "Funding Information" section.

Response: 

We appreciate your attention to detail regarding the Supporting Information. We have included captions for the Supporting Information files at the end of the revised manuscript and have updated the in-text citations accordingly.

#Reviewer 1 comments:

The manuscript "Colon Mucosal Proteomics of Ankylosing Spondylitis versus Gut inflammation" uses proteomic techniques to identify the differentially expressed proteins in the colon tissues of patients with AS and patients with gut inflammation. The authors collected 20 samples from 4 diff groups: healthy individuals (group A), patients with gut inflammation only (group B), patients with AS only (group C), and patients with AS combined with gut inflammation (group D).The authors found differential protein expression between all the groups. Further analysis revealed that the NmrA-like family domain containing 1 (NMRAL1) protein was identified as a DEP specifically associated with group D. The authors performed a knockdown of NMRAL1 using siRNA and checked for the levels of key inflammatory cytokines and found that the levels were significantly decreased. The manuscript is well written and the conclusions are based 

on results but I have a few concerns.

Response:

Thank you very much for your encouraging feedback and for recognizing the strengths of our study. We are grateful for your thoughtful review and the constructive insights you provided. We appreciate your acknowledgment of our manuscript’s clarity and the alignment of our conclusions with our results. We have carefully considered each of your comments and have made the necessary revisions to address your concerns. We hope that the changes we have implemented further strengthen the quality of our 

work and provide a clearer understanding of the findings.

1. The figure quality is really bad and I cannot read anything even after zooming in. Though the authors have provided tiff figures, the figure quality in the manuscript need to be improved.

Response: 

We apologize for the poor quality of the figures in the original submission. To enhance clarity and readability, we have replaced the figures with high-resolution (600 dpi) versions to ensure that all data is clearly visible. The updated figures are now clearer and should be more easily interpretable. Thank you for your understanding.

2. In the figures the authors have used A, B, C, D to demarcate the diff groups. As a reader it is diff to go back and check what each group is. I suggest that the authors use actual group headings instead of A, B, C, D. For example Healthy, Gut Inflammation, AS and AS with Gut inflammation.

Response: 

We appreciate your feedback regarding the group labels in the figures. To make it easier for readers, we have updated the figures to use descriptive labels (e.g., "Healthy," "Gut Inflammation," "AS," and "AS with Gut Inflammation") instead of A, B, C, and D. We hope this change enhances the readability of our data.

3. In the volcano plots, if the authors can mark the names of some of the highly differentially expressed proteins it will make the reading more interesting.

Response: 

Thank you for your advice. We agree that marking the names of some highly differentially expressed proteins in the volcano plots would make the results more engaging. We have revised the plots accordingly, highlighting key differentially expressed proteins to help guide readers through the significant findings. We also described the results of the highly differentially expressed proteins identified as markers in the revised manuscript.

(lines 268-272 in revised manuscript)Among the highly differentially expressed proteins identified, SPATS2L, PCSK1, and FAM98A showed significant upregulation in group B compared to group A. COL5A1 was notably upregulated in group D compared to group C, while APCS and ALPL were significantly different between groups D and B. Additionally, CARHSP1 was identified as highly differentially expressed between groups C and A. These proteins are marked in the volcano plots (Fig 2a, b, c, d). 

4. Please make sure to add all the recent references.

Response: 

Thank you for your valuable feedback. In response to your suggestion to include recent references, we have updated the reference list as follows:

Replacement of Outdated References: We have reviewed the original reference list and replaced older references with more recent studies that reflect current advances in the field. This ensures that our manuscript is up-to-date with the latest findings. 

Addition of New References: We have supplemented the reference list with additional studies from the past three years, covering key areas such as the etiology and pathogenesis of ankylosing spondylitis, gut microbiota interactions, and relevant proteomics research. These references provide further support for our study and highlight recent progress in related research areas. 

The specific modifications are as follows:

Introduction

(lines 57-62 in revised manuscript) The etiology and pathogenesis of AS remain unclear but are thought to be related to factors such as genetics, oxidative stress, mineral metabolism disorders, smoking, infections, and gut microbiota[3]. The gut microbiota is a complex microbial ecosystem that plays a crucial role in the development of various autoimmune diseases, including multiple sclerosis, rheumatoid arthritis, type 1 diabetes and systemic lupus erythematosus[4].

[3] Bilski R, Kamiński P, Kupczyk D, Jeka S, Baszyński J, Tkaczenko H, et al. Environmental and Genetic Determinants of Ankylosing Spondylitis. Int J Mol Sci. 2024;25(14):7814. doi:10.3390/ijms25147814

[4] Miyauchi E, Shimokawa C, Steimle A, Desai MS, Ohno H. The impact of the gut microbiome on extra-intestinal autoimmune diseases. Nat Rev Immunol. 2023;23(1):9-23. doi:10.1038/s41577-022-00727-y

Discussion

(lines 371-374 in revised manuscript) Campos JF etal. recently discovered that patients with spondyloarthritis (SpA) have a high prevalence of microscopic gut inflammation, and fecal calprotectin levels can effectively identify microscopic inflammation, serving as a biomarker for microscopic intestinal inflammation in SpA patients[16].

(lines 376-377 in revised manuscript) Previous studies on AS biomarkers have primarily focused on serum proteomics, lipidomics, and metabolomics[18-20].

(lines 409-413 in revised manuscript) TNF-α is widely recognized as a central inflammatory cytokine in AS. Researchers have found that high concentrations of TNF-α can induce METTL14-mediated m6A modification of the ELMO1 3′ UTR, enhancing the migration of AS mesenchymal stem cells, thereby promoting the progression of AS through pathological osteogenesis[33].

(lines 415-416 in revised manuscript) A mendelian randomization study found that IL-1β, a pro-inflammatory cytokine, plays a key role in the pathogenesis of AS[35].

[16]Campos JF, Resende GG, Barbosa AJA, de Carvalho SC, Lage JA, Cunha PFS, et al. Fecal calprotectin as a biomarker of microscopic bowel inflammation in patients with spondyloarthritis. Int J Rheum Dis. 2022;25(9):1078-1086. doi:10.1111/1756-185X.14388

[18]Hwang M, Assassi S, Zheng J, Castillo J, Chavez R, Vanarsa K, et al. Quantitative proteomic screening uncovers candidate diagnostic and monitoring serum biomarkers of ankylosing spondylitis. Arthritis Res Ther. 2023;25(1):57. doi:10.1186/s13075-023-03044-4

[19]Li Z, Gu W, Wang Y, Qin B, Ji W, Wang Z, et al. Untargeted Lipidomics Reveals Characteristic Biomarkers in Patients with Ankylosing Spondylitis Disease. Biomedicines. 2022;11(1):47. doi:10.3390/biomedicines11010047

[20]Li L, Ding S, Wang W, Yang L, Wilson G, Sa Y, et al. Serum metabolomics reveals the metabolic profile and potential biomarkers of ankylosing spondylitis. Mol Omics. 2024;20(8):505-516. doi:10.1039/d4mo00076e

[33] Xie Z, Yu W, Zheng G, Li J, Cen S, Ye G, et al. TNF-α-mediated m6A modification of ELMO1 triggers directional migration of mesenchymal stem cell in ankylosing spondylitis. Nat Commun. 2021;12(1):5373. doi:10.1038/s41467-021-25710-4

[35]Fang P, Liu X, Qiu Y, Wang Y, Wang D, Zhao J, et al. Exploring causal correlations between inflammatory cytokines and ankylosing spondylitis: a bidirectional mendelian-randomization study. Front Immunol. 2023;14:1285106. doi:10.3389/fimmu.2023.1285106

We trust that these updates improve the manuscript by aligning it with current literature and hope it meets your expectations. Thank you once again for your insightful suggestion.

5. A graphical abstract can be useful for the reader.

Response:

Thank you for your thoughtful suggestion regarding a graphical abstract. We agree that it would be a valuable addition for readers, providing a quick and clear overview of our study’s main findings. We have created a graphical abstract and uploaded it as Fig 1 in the revised manuscript that we hope will enhance the accessibility and appeal of the paper. The note of the graphical abstract is shown below：

(lines 80-87 in revised manuscript) Fig 1. The experimental workflow of the colonic mucosal proteomics study on AS with intestinal inflammation. Colon mucosal tissues were collected from four groups: healthy individuals, patients with gut inflammation, patients with AS and patients with AS combined with gut inflammation. These tissues underwent protein lysis, gel electrophoresis separation, trypsin digestion, and subsequent mass spectrometry analysis. The differentially expressed proteins identified were annotated for their biological functions using bioinformatics tools. NMRAL1 was found to be a differentially expressed protein common to all four groups. In vitro experiments were conducted to explore the impact of NMRAL1 on the expression of inflammatory cytokines TNF-α, IL-1β, IL-17, and IL-23, employing cell transfection methods, RT-qPCR, and ELISA assays.

#Reviewer 2 comments:

The manuscript, Colon Mucosal Proteomics of Ankylosing Spondylitis versus Gut Inflammation, examined the impact of NMRAL1 protein on inflammatory cytokines and identified the differentially expressed proteins (DEPs) in the colon tissues of patients with AS and those with gut inflammation and has established this correlation effectively.

It is a well-researched, experimentally supported, and well-presented paper. The study has an excellent design and a comprehensive literature review.

The lowered NMRAL1 expression in response to LPS-induced inflammation, as well as the lower expression with siRNA transfection, suggests that NMRAL1 may play a role in inflammation regulation. The combined evidence from AS samples and in vitro investigations supports the link between NMRAL1 and inflammation-related functions, presumably specific to AS and gut inflammation.

Response: 

Thank you very much for your encouraging feedback and for recognizing the strengths of our study. We truly appreciate your positive remarks regarding our research design, literature review, and the relevance of NMRAL1 in inflammation regulation. Your acknowledgment of our efforts in establishing the link between NMRAL1 and inflammation in AS and gut inflammation is highly motivating for our team.

Query 1: However, there is a query regarding collecting colonic mucosa specimens from 20 patients only. This sample size is assumed to be too low for conducting research and generalization of results. Since small sample sizes are more susceptible to random variation and may not represent the broader AS population accurately and there are many factors like age, medication, disease duration, and lifestyle which can influence inflammation in both the gut and joints, thus how the statistical data of your results shows confidence.

Response:

Thank you for your thoughtful and constructive comment regarding the sample size used in our study. We completely agree with your observation that a small sample size can introduce variability and may not fully represent the broader AS population. As you mentioned, factors such as age, medication, disease duration, and lifestyle can influence inflammation in both the gut and joints. We carefully considered these factors when designing the study and during patient selection.

The average age of the four groups included in our study was between 47 and 50 years, ensuring that the groups were comparable in terms of age. Additionally, we excluded patients who had used biologic agents within the past year to avoid confounding effects related to medication. While the sample size was limited, we applied rigorous statistical methods to assess the significance of our findings. For example, all label-free data were analyzed using the MaxQuant software (version 1.6.14.0) and subjected to Label-Free Quantification (LFQ) with stringent filtering criteria (Peptide FDR≤0.01 and Protein FDR≤0.01) to ensure the accuracy of the results. Furthermore, we conducted Principal Component Analysis (PCA) (Supplementary Figure S1) to examine the overall variability within and between groups. This analysis helped us better understand the data distribution and the relationships between groups.

Most importantly, we believe the strong correlation observed between NMRAL1 expression and inflammation-related functions, supported by both in vitro data, provides a solid foundation for our findings, despite the smaller sample size.

We 

---

## [Decision Letter · Decision Letter 1]

25 Nov 2024

Colon Mucosal Proteomics of Ankylosing Spondylitis versus Gut inflammation

PONE-D-24-44456R1

Dear Dr. He,

We’re pleased to inform you that your manuscript has been judged scientifically suitable for publication and will be formally accepted for publication once it meets all outstanding technical requirements.

Kind regards,

Sayed Haidar Abbas Raza

Academic Editor

PLOS ONE

Additional Editor Comments (optional):

Reviewers' comments:

Reviewer's Responses to Questions

**Comments to the Author**

1. If the authors have adequately addressed your comments raised in a previous round of review and you feel that this manuscript is now acceptable for publication, you may indicate that here to bypass the “Comments to the Author” section, enter your conflict of interest statement in the “Confidential to Editor” section, and submit your "Accept" recommendation.

Reviewer #1: All comments have been addressed

Reviewer #2: All comments have been addressed

2. Is the manuscript technically sound, and do the data support the conclusions?

Reviewer #1: Yes

Reviewer #2: Yes

3. Has the statistical analysis been performed appropriately and rigorously? 

Reviewer #1: Yes

Reviewer #2: Yes

4. Have the authors made all data underlying the findings in their manuscript fully available?

Reviewer #1: Yes

Reviewer #2: Yes

5. Is the manuscript presented in an intelligible fashion and written in standard English?

Reviewer #1: Yes

Reviewer #2: Yes

6. Review Comments to the Author

Reviewer #1: The authors have adequately addressed all the concerns. They have answered each concern point by point.

Reviewer #2: The authors have satisfactorily replied to all the comments been asked in my previous review and I am satisfied with their response.

7. PLOS authors have the option to publish the peer review history of their article (what does this mean?). If published, this will include your full peer review and any attached files.

Reviewer #1: No

Reviewer #2: **Yes: **Harpreet Kaur, UIPS, Panjab University, Chandigarh, India 160014

---

## [Editor Report · Acceptance letter]

4 Dec 2024

PONE-D-24-44456R1 

PLOS ONE

Dear Dr. He, 

I'm pleased to inform you that your manuscript has been deemed suitable for publication in PLOS ONE. Congratulations! Your manuscript is now being handed over to our production team.

Kind regards, 

on behalf of

Dr. Sayed Haidar Abbas Raza 

Academic Editor

PLOS ONE